YOLOv8-Coal: a coal-rock image recognition method based on improved YOLOv8

http://orcid.org/0009-0000-0713-0738 Wang Wenyu 1
Zhao Yanqin 1 982764511@qq.com
Xue Zhi 2
1 School of Computer and Information Engineering, Heilongjiang University of Science and Technology , Harbin, Heilongjiang , China
2 College of Science, Heilongjiang University of Science and Technology , Harbin, Heilongjiang , China
Kong Xiangjie
Electronic publication date: 2024 Sep 16
Publication date: 2024
Volume: 10
Electronic Location ID: e2313
Received 2024 Mar 3; Accepted 2024 Aug 16
Copyright: © 2024 Wang et al.
Copyright year: 2024
Copyright holder: Wang et al.
License: This is an open access article distributed under the terms of the Creative Commons Attribution License, which permits unrestricted use, distribution, reproduction and adaptation in any medium and for any purpose provided that it is properly attributed. For attribution, the original author(s), title, publication source (PeerJ Computer Science) and either DOI or URL of the article must be cited.
License URL: https://creativecommons.org/licenses/by/4.0/

Keywords: Attention mechanisms, Coal-rock image recognition, Deep learning, Object detection, YOLOv8

Funding: Basic Research Operating Costs of Undergraduate Colleges and Universities in Heilongjiang Province Project 2022-KYYWF-0565 Heilongjiang University of Science and Technology 2024 College Students’ Innovation and Entrepreneurship Training Program Project This work was supported by the Basic Research Operating Costs of Undergraduate Colleges and Universities in Heilongjiang Province Project (No.2022-KYYWF-0565) and the Heilongjiang University of Science and Technology 2024 College Students’ Innovation and Entrepreneurship Training Program Project. The funders had no role in study design, data collection and analysis, decision to publish, or preparation of the manuscript.

==============================
To address issues such as misdetection and omission due to low light, image defocus, and worker occlusion in coal-rock image recognition, a new method called YOLOv8-Coal, based on YOLOv8, is introduced to enhance recognition accuracy and processing speed. The Deformable Convolution Network version 3 enhances object feature extraction by adjusting sampling positions with offsets and aligning them closely with the object’s shape. The Polarized Self-Attention module in the feature fusion network emphasizes crucial features and suppresses unnecessary information to minimize irrelevant factors. Additionally, the lightweight C2fGhost module combines the strengths of GhostNet and the C2f module, further decreasing model parameters and computational load. The empirical findings indicate that YOLOv8-Coal has achieved substantial enhancements in all metrics on the coal rock image dataset. More precisely, the values for AP50, AP50:95, and AR50:95 were improved to 77.7%, 62.8%, and 75.0% respectively. In addition, optimal localization recall precision (oLRP) were decreased to 45.6%. In addition, the model parameters were decreased to 2.59M and the FLOPs were reduced to 6.9G. Finally, the size of the model weight file is a mere 5.2 MB. The enhanced algorithm’s advantage is further demonstrated when compared to other commonly used algorithms.

Introduction

Coal is the most plentiful fossil fuel on the planet and is essential for the world’s energy production. Although the proportion of renewable energy is slowly growing, coal continues to be the primary source of power in numerous countries (Dai & Finkelman, 2018). The extraction and utilization of coal impact energy security, environmental protection, and sustainable development goals. Hence, enhancing the efficiency of coal resource development and exploitation while minimizing environmental pollution has emerged as a pressing global issue. The research and implementation of coal-rock image recognition technology are crucial in enhancing mining efficiency, minimizing resource wastage, and reducing environmental interference and damage through the precise identification of coal and rock. Additionally, it facilitates the advancement of intelligent coal mining, as well as technological innovation and sustainable development in the coal industry. Hence, it is essential to research and enhance the coal-rock image identification method.

Prior to the widespread adoption of deep learning in image recognition, prior approaches to coal-rock image detection primarily involved human feature extraction or machine learning algorithms. Sun & Su (2013) utilized gray level co-occurrence matrix to extract 22 texture characteristics from coal rock images. From these features, they manually selected four dimensions and employed the fisher discriminant method for coal rock image classification. Their approach achieved an average recognition rate of 94.12%. Sun & Chen (2015) employed an asymmetric generalized Gaussian model in the wavelet domain, along with an enhanced relative entropy similarity measure, to classify coal-rock images. The technique achieved an average identification rate of 87.77%. However, it is worth noting that the system is susceptible to noise. Wu & Tian (2016) employed a dictionary learning technique to extract characteristics from coal-rock images. They then integrated this with a KNN classification algorithm to accurately categorize and recognize coal rock images, achieving a recognition rate of 96.154%. However, this approach is computationally expensive and has weak generalization capabilities. Wang & Zhang (2020) employed the Local Binary Pattern (LBP) algorithm to analyze variations in coal-rock texture. They retrieved four specific texture features—energy, entropy value, contrast, and inverse differential moment—by utilizing the gray scale covariance matrix of coal rock pictures through GLCM. The findings indicate that the LBP algorithm is highly effective in detecting changes in coal rock features, albeit it lacks robustness. Zhang et al. (2022) employed Gaussian filtering to manipulate the grayscale image of coal gangue and utilized the least squares vector mechanism to construct a coal gangue recognition model. The gangue image achieved recognition accuracies of 92.2% and 91.5% when employing grayscale bias and texture contrast as indicators, respectively. Although manual feature extraction methods have shown initial success, they are not easily scalable or objective. They struggle to adjust to image changes, require redesigning when task requirements change, and are influenced by personal experience and preference. On the other hand, machine learning algorithms face challenges such as overfitting, complex parameter optimization, and limitations in processing complex images.

Deep learning has led to the widespread adoption of convolutional neural networks (CNNs) in the realm of image recognition. CNN can autonomously acquire and isolate visual characteristics, significantly enhancing the precision and effectiveness of image identification. Huiling & Xin (2019) employed a wavelet transform for feature extraction from coal-rock photos and integrated it with a BP neural network for coal-rock identification. They achieved a recognition rate of 96.67% with 10 hidden layer nodes and 500 iterations. Liu et al. (2019) employed a multi-scale Completed Local Binary Pattern (CLBP) and CNN-based Deep Feature Extraction (DFE) technique to recognize coal-rock images. This method combines feature vectors of various scales and feeds them into a nearest neighbor classifier using cardinality distance, resulting in an accuracy of 97.9167%. Pu et al. (2019) employed a CNN-based VGG16 network to identify coal and gangue images. They also introduced the concept of migration learning, achieving a recognition accuracy of 82.5%. Si et al. (2020) enhanced the performance of a CNN for coal-rock detection by incorporating dropout, weight regularization, and batch normalization techniques. The modified CNN achieved an F1-score of 78.62%.

Due to technological advancements and increased demand, basic classification is insufficient for precise recognition and localization in intricate environments. This has resulted in the creation of target detection technology, which must not only identify the object in the image but also properly determine its position. The concept of two-stage detection methods, exemplified by Faster R-CNN (Ren et al., 2016) and Mask R-CNN (He et al., 2018), emerged. These methods initially generate Regions of Interest (RoIs) and subsequently classify and regress the bounding boxes of these regions. Hua, Xing & Zhao (2019) employed Faster R-CNN to identify coal-rock. They extracted features using the VGG16 network and obtained the coordinates of coal seam boundaries using the R-CNN network. The mAP achieved was approximately 88%. Shan et al. (2022) enhanced the ResNet50 feature extraction network of Faster R-CNN by integrating the attention mechanism algorithm CBAM. They applied this improved model to identify mixed and release state of coal-gangue masses of fully mechanized caving. The average detection and recall rates of the improved model achieved 82.63% and 86.53% respectively, resulting in a 7% improvement in the F1-score. Cao et al. (2024) enhanced the Mask R-CNN network specifically for coal and gangue segmentation. They introduced the Multichannel Forward-Linked Confusion Convolution Module (MFCCM) as a means to develop feature extraction networks. A structure called multiscale high-resolution feature pyramid network is created, together with a proposed structure called multiscale mask head structure. The enhanced algorithm achieves a precision of 97.38%, surpassing the original model by 1.66%.

Two-stage detection approaches often excel in accuracy for target detection but suffer from large computing demands and sluggish processing speed, making them unsuitable for scenarios requiring high real-time performance. Single-stage detection methods predict object bounding boxes and categories directly in the image without step-by-step target classification and positional regression. This simplifies feature processing, reduces computational complexity, and improves processing speed. Examples of such methods are SSD (Liu et al., 2016) and YOLO (Redmon et al., 2016). Zhang et al. (2020) utilized the YOLOv2 algorithm for the identification of underground coal and rock images, achieving a recognition accuracy of 78% and a detection speed of 63 frame/s. The accuracy of this method surpassed that of Faster R-CNN and SSD by 7.7% and 4.7%, respectively, while the detection speed exceeded these models by 763% and 40%, respectively. Sun et al. (2022) enhanced the YOLOv3 technique by using depth-separable convolution and applied the cubic spline interpolation algorithm to accurately adjust the bounding box. The enhanced YOLOv3 algorithm achieved a 5.85% increase in accuracy in the x direction and a 16.99% increase in accuracy in the y direction. Additionally, it reduced the number of parameters by approximately 80% and decreased the prediction time by around 5%. Li et al. (2022) introduced a YOLOv3 technique that utilizes deformable convolutions to detect gangue. They improved the accuracy of anchor frame localization by averaging the outcomes of repeated k-means clustering. This approach resulted in a mAP of 99.45% and a 61.4% reduction in the maximum FLOPs required. Liu et al. (2021a) suggested an enhancement to YOLOv4 for the purpose of identifying coal and gangue. Cluster analysis was used to optimize the anchor values on various datasets. Additionally, the number of layers in the feature pyramid was increased. As a result, the mAP of the enhanced network improved by 0.81%. Wang et al. (2022) enhanced the YOLOv5 algorithm by incorporating the CBAM attention mechanism and Transformer into the network for the purpose of identifying coal rock images. The enhanced YOLOv5 network achieves a 92.8% mAP, however, the inclusion of CBAM introduces significant computational complexity and excessive dependence on global information, resulting in the model disregarding crucial local properties. Zhao & Deng (2024) enhanced the YOLOv7 network by substituting a portion of the CBS structure with the ConvNext module, which utilizes 7 × 7 convolutional kernels. Additionally, they included the SimAM attention mechanism along with the αIoU loss function to accurately identify coal rock images. The enhanced model demonstrated a 3.9% increase in accuracy and a 1.5% increase in mAP. This suggests that incorporating the attention mechanism with convolution modification can enhance the effectiveness of the model. However, the use of larger convolution kernels is computationally demanding and fails to effectively capture subtle features. Although these models have shown some success, they still have shortcomings, such as inadequate precision, when handling complex coal-rock pictures.

The YOLO algorithm has been a significant breakthrough in target detection since its introduction, experiencing numerous revisions and enhancements to achieve higher levels of performance and efficiency. YOLOv8 (Ultralytics, 2023) enhances accuracy while optimising speed and model size, making it particularly well-suited for coal-rock image recognition tasks. In response to the issues of poor accuracy and slow processing speeds encountered by various methods when handling complex coal rock images, a coal rock image recognition method based on an improved YOLOv8 algorithm, termed YOLOv8-Coal, has been proposed. The method enhances the recognition capability of coal-rock by modifying the model structure, thereby optimising the feature processing process. Additionally, it reduces the computational complexity of the model, improves processing speed, and better accommodates real-time detection requirements. This article’s primary contributions are: (1) Deformable Convolutional Networks Version 3 (DCNv3) has been introduced to enhance its ability to adjust to the specific features of an image by incorporating dynamically deformable convolution kernels. This allows for better representation of irregular shapes and intricate textures in coal-rock images, ultimately leading to improved model accuracy.

(2) Polarized Self-Attention (PSA) is incorporated to pinpoint and emphasize crucial aspects of the features while diminishing attention towards less significant information. This allows the model to concentrate on the essential elements in the image, thereby enhancing its performance.

(3) The GhostNet concept is integrated into the C2f module to create the C2fGhost module, which enhances the model’s performance, reduces its need for computer resources, and optimizes its computational efficiency and response time.

The rest of the article is organized as follows: The “Materials and Methods” section presents the YOLOv8 model and the enhanced YOLOv8-Coal model principle. The “Experiments, Results and Discussion” section examines the proposed method’s application and provides the experimental results. Lastly, the “Conclusions” section summarizes the entire work and discusses future development directions.

Materials and Methods

This section will provide a quick introduction to the YOLOv8 algorithm, followed by the presentation of the enhanced YOLOv8-Coal algorithm. A set of structural adjustments will be made to enhance the model’s ability to recognize coal-rock effectively.

Overview of the YOLOv8 algorithm

Ultralytics has introduced YOLOv8, which comprises five models: YOLOv8n, YOLOv8s, YOLOv8m, YOLOv8l, and YOLOv8x. These models vary in terms of network depth and feature map size. YOLOv8n stands out for its exceptional processing speed, whilst YOLOv8x excels in terms of detection accuracy. This study selects YOLOv8n as the benchmark model for enhancement, with the goal of maintaining high detection accuracy while maximizing computational efficiency.

The YOLOv8 network consists of four main components: Input, Backbone, Neck, and Head. The Input utilizes a mosaic data augmentation strategy, significantly enhancing the model’s adaptability to scale variations and complex backgrounds. The Backbone introduces the Faster Implementation of CSP Bottleneck with two convolutions (C2f) and Spatial Pyramid Pooling-Fast (SPPF) modules. The C2f module enhances the representation of complex features by adding more branches and cross-layer connections. SPPF incorporates a rapid pooling operation, allowing for the input of images of arbitrary sizes and improving feature extraction capabilities. The Neck features a Path Aggregation Network with Feature Pyramid Network (PAFPN) structure, based on the C2f module, which integrates the top-down feature fusion of Feature Pyramid Networks (FPN) and the bottom-up feature enhancement of Path Aggregation Network (PAN) for more efficient feature extraction. The Head employs a Decoupled-Head and Anchor-Free strategy. The Decoupled-Head separates the classification and regression tasks, enabling independent learning in the classification branch from the regression branch. The Anchor-Free strategy directly predicts the centers of objects, eliminating the dependency on anchor box sizes and ratios, thereby enhancing the model’s generalization capabilities.

YOLOv8-Coal algorithm

Although YOLOv8 is effective in several situations as a versatile target detection model, it encounters certain difficulties when directly used for coal-rock image recognition. The challenges arise from the distinct features of these images. Firstly, there is a significant resemblance in texture between coal and rock, which poses a considerable challenge for the model’s ability to differentiate between them. Secondly, the underground environment is characterised by varying lighting conditions, including fluctuating light intensity and different types of light sources. This makes it difficult to accurately represent the visual information captured in images taken underground in the mine complex. Furthermore, given the constraints of the equipment used in underground coal mines, the model must not only achieve a high level of recognition accuracy, but also minimise the utilisation of computational resources. Hence, in this research, YOLOv8 is precisely fine-tuned to improve its capacity in identifying coal-rock images and maintaining a lightweight model, thereby achieving efficient and precise image recognition in the underground setting.

Aiming at the special needs of the coal rock image recognition task, this study carried out an in-depth optimization of the YOLOv8 model, innovatively integrated several advanced modules, and proposed a coal rock image recognition model, which was named YOLOv8-Coal, and its structure is shown in Fig. 1. First, to address the issue of the strong resemblance between coal and rock textures in coal-rock images, the C2f module in the feature fusion network is substituted with the Deformable Convolution Network version 3 (DCNv3) module (Wang et al., 2023a). The DCNv3 module employs a deformable convolution kernel, which allows it to better accommodate the non-linearities and irregular variations present in the image. This is especially beneficial for distinguishing the subtle distinctions between coal and rock, and it further improves the model’s capability to extract features. Second, to improve the effectiveness of capturing key information in coal rock images, we have added the PSA module (Liu et al., 2021b) after the last DCNv3 module and before the three detection heads. This module enhances the identification of important features in the coal rock by dynamically adjusting the network’s focus. It helps the model identify crucial features in the coal rock even in challenging conditions such as uneven illumination and occlusion. By focusing on these decisive features, the model’s accuracy in recognizing the coal-rock interface is improved, and its robustness is greatly enhanced across different lighting and environmental conditions. Finally, the introduction of a lightweight C2fGhost module (Han et al., 2020) replaces the original C2f module in the backbone network. This replacement reduces feature redundancy while keeping the original features intact. As a result, the number of parameters and computation in the model is further reduced, making it more suitable for operation in resource-constrained coal mine underground environments. The YOLOv8-Coal model showcases exceptional performance in tackling the unique issues associated with coal rock image recognition through its methodical reconfiguration and integration strategy. The integration of these modules not only enhances the accuracy and efficiency of recognition, but also guarantees their suitability and effectiveness in resource-limited underground coal mine environments. This highlights the innovative contribution of this research in the field of coal rock image recognition.

Figure 1 YOLOv8-Coal network structure.

DCNv3 module

This study introduces the DCNv3 module as a means of enhancing the accuracy of coal rock image recognition in YOLOv8. The conventional convolution technique is constrained by the fixed sample position and exhibits limited generalization capability. DCNv3 is a network architecture that utilizes deformable convolution to allow the network to flexibly modify the shape and size of the convolution kernel. This adaptation helps the network better accommodate the specific characteristics of the image’s local features. The introduction of dynamically deformable convolution kernels enables this functionality, as illustrated in Fig. 2.

Figure 2 Principle of deformable convolution network version 3.

DCNv3 introduces an offset to the standard convolution’s sampling position, allowing the grid to deform freely. This offset can be adjusted through backpropagation, resulting in a sensing field that closely matches the object’s shape. This enables more accurate feature extraction, as demonstrated in Eq. (1).

(1) y(p0)=∑g=1G∑k=1Kwgmgkxg(p0+pk+Δpgk)

where G is defined as the total number of aggregated groups, and K represents the number of sampling points within each aggregation group. Specifically, for the g-th group, the dimension of the group is denoted as C′=C/G, and the location-irrelevant projection weights of the group is denoted as wg∈RC×C′. The modulation scalar of the k-th sampling point in the g-th group is mgk∈R, which is normalized by a Softmax function along dimension K. The sliced input feature map is denoted as xg∈RC′×H×W. p0 represents the current pixel provided, while Δpgk represents the offset that corresponds to the grid sampling position pk of the gth group.

DCNv3 features weight sharing among convolutional neurons, a multi-grouping mechanism, and a modulation factor generating approach. Sharing weights among convolutional neurons draws on the idea of separable convolutions, separating the original convolutional weights into two parts: channel-by-channel convolution and point-by-point convolution, so that its parameter and storage complexity are no longer linearly related to the total number of sampling points, which improves the efficiency of the model; the multi-grouping mechanism is to divide the spatial aggregation process into multiple groups, each of which has independent sampling offsets and modulation scales, so that different groups can have different spatial aggregation patterns on the convolution layer, which can better capture the changing patterns of the target at different locations and scales; and the modulation factor generation method is to replace the Sigmoid normalization based on bit-by-bit operation with Softmax normalization based on sample points, so that the sum of modulation scalars is restricted to 1, which makes the training process of the model more stable at different scales.

The DCNv3 module is well-suited for addressing the intricate texture and irregular shape issues that are frequently encountered in coal rock photos. This module increases the model’s flexibility and recognition accuracy for coal-rock-specific textures by using a dynamically changeable convolutional kernel. Additionally, it enhances the model’s generalization capacity in a variable environment. DCNv3 is very good in accurately distinguishing between coal and rock. It can effectively address the challenges of classifying these materials, which often have similar textures. As a result, DCNv3 greatly enhances the accuracy and efficiency of recognition. Hence, the implementation of DCNv3 not only improves the efficiency of the current model architecture, but also introduces a novel approach for the targeted job of recognizing coal-rock images, thereby enhancing the model’s performance in real-world scenarios.

PSA module

To enhance the accuracy of identifying and locating the important characteristics in the image of the coal rock, we have introduced the PSA module. The structure of this module is illustrated in Fig. 3. The PSA is a sophisticated attention mechanism that improves the model’s ability to concentrate on important aspects by using polarization operations. It also reduces the impact of redundant information, allowing the model to better acquire and utilize crucial information.

Figure 3 Polarized Self-Attention structures in parallel layout.

The PSA is bifurcated into two branches: the channel dimension and the spatial dimension. Equation (2) displays the formula used to assign weight to the channel dimension.

(2) Ach(X)=FSG[Wz|θ1((σ1(Wv(X))×FSM(σ2(Wq(X)))))]

where Ach(X)∈RC×1×1, X represents the input feature tensor, Wq, Wv and Wz denote 1 × 1 convolution layers, θ1 is an intermediate parameter for the convolution of these channels, σ1 and σ2 are two dimensionality reduction operations, FSM(⋅) represents the Softmax function, FSM(X)=∑j=1Npexj∑m=1Np⁡exmxj, FSG(⋅) represents the Sigmoid function, “ ×” means matrix dot product operation, and the number of internal channels between Wv∣Wq and Wz is C/2. Then the output of the channel dimension is Zch=Ach(X)⊙chX∈RC×H×W, where ⊙ch denotes a channel-wise multiplication operator.

Equation (3) displays the formula used to assign weight to the spatial dimension.

(3) Asp(X)=FSG[σ3(FSM(σ1(FGP(Wq(X))))×σ2(Wv(X)))]

where Asp(X)∈R1×H×W, X represents the input feature tensor, Wq and Wv denote 1 × 1 convolution layers, σ1 and σ2 are two dimensionality reduction operations, σ3 is an ascending operation, FSG(⋅), FSM(⋅) and FGP(⋅) stand for the Softmax function, the Sigmoid function, and the global pooling respectively, FGP(X)=1H×W∑i=1H⁡∑j=1W⁡X(:,i,j), “ ×” means matrix dot product operation. Then the output of the spatial dimension is Zsp=Asp(X)⊙spX∈RC×H×W, where ⊙sp denotes a spatial-wise multiplication operator.

The outputs of these two branches are fused in parallel to obtain the output of the structure as shown in Eq. (4).

(4) PSAp(X)=Zch+Zsp=Ach(X)⊙chX+Asp(X)⊙spX

where “ +” is the element-by-element addition operator.

The PSA mechanism integrates spatial and channel dimensions to achieve precise pixel-by-pixel regression using polarization filtering and High Dynamic Range (HDR), while preserving high resolution in both dimensions. Polarization filtering compresses features in one dimension while preserving high resolution in the perpendicular dimension, minimizing information loss from downscaling. HDR enhances the dynamic range of attention by applying Softmax normalization to the minimum tensor in the attention module, followed by projection mapping using a Sigmoid function to ensure all parameters are within the range of 0 to 1.

By incorporating the PSA module into the feature fusion network, it becomes possible to assign different weights to features and merge them at various levels. This mechanism effectively mitigates the impact of background noise and irrelevant information by enhancing the focus on crucial characteristics in coal rock images. As a result, the model’s recognition efficiency and accuracy are significantly improved. The PSA module is particularly effective when working with coal rock images that have intricate textures and irregular shapes. It ensures that the network utilizes the feature information from each layer to its fullest extent, thereby optimizing the overall process of extracting and recognizing features. The implementation of this focused feature processing strategy allows the model to achieve greater accuracy and resilience in recognizing coal rock images, particularly in situations with uneven lighting and visual obstruction.

C2fGhost module

C2fGhost is employed as an enhanced lightweight module to minimise model computation without compromising accuracy. Deep neural networks typically include feature maps that contain abundant information, but often exhibit redundancy due to the presence of numerous pairs of comparable feature maps. The Ghost module utilises cost-effective linear operations and produces intrinsic feature maps to enhance the generation of feature maps. This approach allows for the creation of feature maps with reduced computational requirements and fewer parameters, resulting in lower computational costs without compromising performance.

The distinction between regular convolution and the Ghost module is illustrated in Fig. 4. During the process of forward propagation, the Ghost module initially creates an output feature map by regular convolution. Subsequently, this feature map is utilized to construct a second feature map through cost-effective linear operations. Ultimately, the two feature maps are combined in the channel dimension to provide the ultimate result. The Ghost module is more efficient than regular convolution as it decreases the number of parameters and computations while maintaining the same size of the output feature maps.

Figure 4 The ordinary convolution and the Ghost module.

Assume that the input data is h×w×c, the output data is h′×w′×n, and the size of the convolution kernel is k×k, where h and w are the height and width of the input data, h′ and w′ are the height and width of the output data, and c and n are the number of input and output channels, respectively. Then the computation of ordinary convolution is n×h′×w′×c×k×k. The Ghost module is able to generate the same number of feature maps as the ordinary convolution using fewer operations. Assuming that d×d is the kernel size of the linear operation and it is of similar order of magnitude to k×k, and s denotes the number of cheap linear operations and s≪c, the computational amount of the Ghost module is ns×h′×w′×c×k×k+(s−1)×ns×h′×w′×d×d. Thus, the conventional convolution is substituted with the Ghost module, which exhibits a theoretical speed-up ratio, as depicted in Eq. (5).

(5) rs=n⋅h′⋅w′⋅c⋅k⋅kns⋅h′⋅w′⋅c⋅k⋅k+(s−1)⋅ns⋅h′⋅w′⋅d⋅d=c⋅k⋅k1s⋅c⋅k⋅k+s−1s⋅d⋅d≈s⋅cs+c−1≈s.

As can be seen from Eq. (5), the computation of the ordinary convolution is approximately s times that of the Ghost module. The parametric quantity is computed by a similar process, again approximated by s.

GhostBottleneck was introduced as an extension of Ghost modules, incorporating a shortcut connection and two stacked Ghost modules. The initial Ghost module is employed to augment the quantity of channels, whilst the subsequent module is utilized to diminish the amount of channels in order to align with the shortcuts. Shortcuts are employed to connect the inputs and outputs of the two modules in order to generate the output of the GhostBottleneck. The C2fGhost module is utilized as a replacement for the Bottleneck in the C2f module. The C2fGhost module incorporates the benefits of the Ghost module while reducing the number of parameters and processing requirements. This module offers an efficient solution for coal-rock identification, particularly in situations with limited computational resources, as depicted in Fig. 5.

Figure 5 Ghost module, GhostBottleneck and C2fGhost.

The incorporation of the C2fGhost module into the backbone network significantly decreases the parameters and performance demands of the model, allowing the deep learning model to efficiently operate on resource-limited end devices. The module employs a lightweight feature extraction strategy while analysing photos of coal rock. This approach not only preserves the capacity to extract features with high precision, but also enhances the model’s suitability and usefulness in real-world application scenarios.

Experiments, results and discussion

The enhanced YOLOv8-Coal model is assessed for its effectiveness by testing it with the coal-rock dataset. This section will provide a detailed description of the coal-rock dataset, experimental setup, parameter settings, and model evaluation criteria. Subsequent experiments will showcase the effectiveness of the proposed approach.

Experiment

Coal-rock dataset

The dataset utilized in this study is sourced from the School of Mining Engineering at Heilongjiang University of Science and Technology. It comprises 70 images obtained from two distinct coal mines, as depicted in Fig. 6. The two scenes were captured using an Olympus TG-320 and a Huawei PCT-AL10 with resolutions of 4,288 × 3,216 and 4,000 × 3,000, respectively. Coals typically have a darker color and sparse texture, while rocks have a lighter color and dense texture, making them easily distinguishable. Recognizing coal in practice can be challenging due to various factors, such as image defocus from camera limitations, coal being partially obscured by workers, similar colors between coal and rock in dim mine lighting, and color changes in coal and rock from water exposure. Several image examples are displayed in Fig. 7. To enhance the precision of coal mining operations and reduce the chances of errors, a prudent approach is taken by deliberately disregarding areas that are challenging to identify due to image quality issues like poor lighting and image blurriness during image annotation. This action aims to guarantee the dataset’s quality, thereby enhancing the reliability of the coal identification process.

Figure 6 (A and B) Images from different coal mine scenes.

Image credit: Wang et al. (2024). Coal and Rock [Data set]. Zenodo. https://doi.org/10.5281/zenodo.10702704.

Figure 7 Demonstration of the difficulty of recognizing coal-rock images.

Image credit: Wang et al. (2024). Coal and Rock [Data set]. Zenodo. https://doi.org/10.5281/zenodo.10702704.

Step 1: Image acquisition and pre-processing. A double-three interpolation algorithm was utilized to adjust the resolution of an image from 4,288 × 3,216 to 4,000 × 3,000, addressing the issue of inconsistent resolution from various acquisition devices. Afterwards, each image was divided into 12 non-overlapping 1,000 × 1,000 blocks. This was achieved by cropping the image four times horizontally and three times vertically, creating a total of 840 blocks. In 527 blocks containing coal with specific targets were chosen through manual filtering. The coals in the images were labeled using the LabelImg tool, and a total of 813 coal targets were identified. The study exclusively focused on annotating images of a specific target category, namely “coal”, and did not include any other categories. By focusing on a single target category, this method simplifies the training and validation processes of the model.

Step 2: Dataset segmentation. The blocks labeled above are divided into three sets for different purposes. The training set consists of 70% (368 images) and is used to train the deep learning model. This set provides enough samples for the model to learn the complex features in the data. The validation set, which comprises 20% (106 images), is used for model tuning and to prevent overfitting. Finally, the test set consists of the remaining 10% (53 images) and is used to independently evaluate the performance of the final model. The data partitioning strategy described here has been extensively utilized in numerous deep learning research studies and has demonstrated its efficacy in achieving a balance between model training and evaluation requirements (Faghani et al., 2022; Humayun, Bhatti & Khurshid, 2023). Furthermore, this allocation adheres to commonly employed methodologies in numerous studies, facilitating straightforward comparisons with other research.

Step 3: Image enhancements. To ensure accurate coal-rock identification, a significant amount of data is required. Offline data enhancement through image panning was utilized to transform the images into multiple 640 × 640 images with overlapping sections. Beginning from the upper left corner pixel of every 1,000 × 1,000 block, cropping was done horizontally at intervals of 180 pixels, allowing for a total of three cropping actions. Next, 180 pixels are scrolled vertically and cropping proceeds horizontally, repeating the process. Ultimately, nine 640 × 640 images were acquired from each 1,000 × 1,000 block following data enhancement. By enhancing the images, it can simulate the distribution of coal from various shooting angles, hence improving the model’s recognition performance in different situations and strengthening its robustness.

The final enhanced dataset comprises 4,743 images, consisting of 3,312 training images, 954 validation images, and 477 test images. Figure 8 illustrates the sequence of steps involved in processing the dataset, while Table 1 provides the specific quantities of images and samples at each stage of the dataset.

Figure 8 Dataset processing flow.

Image credit: Wang et al. (2024). Coal and Rock [Data set]. Zenodo. https://doi.org/10.5281/zenodo.10702704.

Table 1 Number of images and samples at different stages of the dataset.

Stage	Total images/samples	Training set images/samples	Validation set images/samples	Test set images/samples	
Initial state	70/0	0/0	0/0	0/0	
Split	840/0	0/0	0/0	0/0	
Filtered & annotated	527/813	368/587	106/150	53/76	
Enhanced	4,743/6,249	3,312/4,460	954/1,191	477/598	

Experimental environment and parameter configuration

The study’s model training and evaluation were conducted on a high-performance server featuring an Intel(R) Xeon(R) Platinum 8358P @ 2.60 GHz processor, 80 GB of RAM, and an NVIDIA GeForce RTX 3090 24 GB graphics card. Furthermore, the server operates on the Ubuntu 20.04 LTS operating system, equipped with CUDA 11.8 and CuDNN 8.4.0. The software utilizes Python 3.8 as the runtime environment and depends on the PyTorch 2.0.0 deep learning framework, along with torchvision 0.15.2 for image processing. It also incorporates the Ultralytics 8.0.195, mmcv 2.0.0, pillow 10.2.0, thop 0.1.1, and pycocotools 2.0.7 libraries to enhance its capabilities in image processing and performance analysis.

The configuration of parameters for this experiment is as follows: The epochs is set to 300 rounds, based on an analysis of experiment outcomes ranging from 100 to 500 rounds. This selection ensures that the model learns enough and prevents overfitting. The patience was chosen at 50 rounds so as to maximise the efficiency of computational resources and avoid overfitting. The batch was determined as 32 in order to consider the specific attributes of the dataset and the constraints of the computational resources. According to Masters & Luschi (2018), this enhances the model’s performance and ensures the training process remains stable. Imgsz was determined to be 640 pixels, taking into account the characteristics of the dataset and the dimensions of the data-enhanced image (640 × 640). Workers was set to 15 in order to minimise data loading time and maximise the utilisation of parallel processing capabilities. The optimizer SGD was selected with an initial learning rate (lr0) and final learning rate (lrf) of 0.01, momentum of 0.937, and weight_decay of 0.0005. These parameter values adhere to the recommendations of Ultralytics, which have been proven to yield outstanding training outcomes and ensure model stability.

Evaluation metrics

In order to provide a comprehensive measure of the performance of the coal-rock recognition model, the following metrics are used to evaluate it: precision, recall, average precision (AP50, AP50:95), average recall (AR50:95), optimal localization recall precision (oLRP), the predicted bounding box count (BBox Count), number of parameters (Params), number of floating point operations (FLOPs), and model weight size (Weight).

Assuming that TP represent the number of correctly classified positive cases, FP represent the number of misclassified negative cases, FN represent the number of misclassified positive cases, and TN represent the number of correctly classified negative cases. Precision refers to the probability of truly being a positive class among all the samples predicted to be positive classes, which is defined as shown in Eq. (6):

(6) Precision=TPTP+FP.

Recall refers to the probability of being predicted as a positive class in a sample that is actually a positive class and is defined as shown in Eq. (7):

(7) Recall=TPTP+FN.

Given that this study focuses on only one specific category (coal), we computed the Average Precision (AP) for this category to assess the model’s detection performance, rather than using the mean Average Precision (mAP). Equation (8) provides the precise definition of average precision.

(8) AP=∫01P(R)dR

where the notation P(R) represents the precision when the recall ratio is R. AP50 refers to the average precision calculated when the Intersection over Union (IoU) ratio is set to 0.5. AP50:95 represents the average precision calculated over various IoU thresholds ranging from 0.50 to 0.95, with a step size of 0.05.

The Average Recall (AR) is a metric that computes the mean value of the model’s recall at various IoU thresholds. It is mathematically defined by Eq. (9).

(9) AR=1NIoU∑t=IoUminIoUmaxRt

where the IoU threshold ranges from IoUmin to IoUmax with an increment of ΔIoU. NIoU represents the number of IoU thresholds used to calculate the recall rate. It is calculated using the formula NIoU=IoUmax−IoUminΔIoU+1. Rt represents the recall at a specific IoU threshold t. The notation AR50:95 represents the mean average precision across various IoU thresholds, ranging from 0.50 to 0.95 with a step size of 0.05.

Localization-Recall-Precision (LRP) (Oksuz et al., 2021) is a comprehensive error metric that considers both the accuracy of localization and classification. The LRP error is defined according to Eq. (10):

(10) LRP(G,D)=1Z(∑i=1NTP1−lq(gi,di)1−τ+NFP+NFN)

where D represents a collection of detection sets that include information about the category and location, while G represents a collection of real labeling sets. Prior to computation, the detection item di∈D is allocated to the appropriate true marker item gi∈G based on the matching criteria of the IoU. NTP, NFP, and NFN denote the quantities of true positive examples, false positive examples, and false negative examples, respectively. The function lq(gi,di) denotes the locational quality of the genuine instances, where di is the actual marker that corresponds to the true marker gi in the genuine instances. Z=NTP+NFP+NFN is the normalization constant. The true example validation threshold, denoted as τ, is set to 0.50.

The Optimal Localization Recall Precision (oLRP) is the minimum possible error in LRP that can be achieved within the detection threshold, or in other words, the confidence score. The confidence score is defined by Eq. (11).

(11) oLRP:=mins∈SLRP(G,Ds)

where the set D& consists of detections that meet or exceed a confidence score threshold &. Detections with confidence scores below & are discarded. The formula involves searching through a set of confidence scores S to determine the optimal trade-off between precision, recall, and localization error. A lower value of oLRP indicates superior performance of the model.

The predicted bounding box count (BBox Count) refers to the overall number of bounding boxes that the model has identified as containing the target object. Params represent the total number of trainable parameters in the model, used to quantify the spatial intricacy of the model. FLOPs are utilized for determining the time complexity, enabling the measurement of the model’s complexity. The term “model weight size” denotes the size of the weight file generated and saved upon completion of the final training process.

Results and discussion

Ablation experiments

The ablation experiments employ the control variable method to confirm how various enhancements affect the model’s performance. For this study, eight sets of experiments were conducted using the same data sets and parameters for each set. The results of the experiments are presented in Table 2. The baseline model refers to the original YOLOv8n network without any additional modules, identified as serial number 1. Following that, three distinct modules for enhancing the network, namely DCNv3, PSA, and C2fGhost, were added to the original model. This resulted in the creation of three new networks: Baseline+DCNv3, Baseline+PSA, and Baseline+C2fGhost. These networks are identified by the serial numbers 2, 3, and 4, respectively. Furthermore, the study examined the impact of integrating these modules to create three two-module enhancement networks: Baseline+DCNv3+PSA, Baseline+DCNv3+C2fGhost, and Baseline+PSA+C2fGhost, denoted as serial numbers 5, 6, and 7, respectively. Finally, three modules were integrated into a single network, forming Baseline+DCNv3+PSA+C2fGhost, which is the YOLOv8-Coal proposed in this study. This integration was done to evaluate the synergistic effects of multiple modules on enhancing network performance, corresponding to serial number 8. Figure 9 displays the various performance metrics of YOLOv8-Coal during both training and validation.

Table 2 Results of the ablation experiments.

Bold entries represent the highest values for each metric.

NO.	DCN
v3	PSA	C2f-Ghost	Precision
(%)	Recall
(%)	AP50
(%)	AP50:95
(%)	AR50:95
(%)	oLRP
(%)	BBox Count	Params
(M)	FLOPs
(G)	Weight
(MB)	
1	×	×	×	90.7	68.2	74.7	59.5	73.6	47.1	25,555	3.01	8.2	6.0	
2	✓	×	×	90.1	66.4	74.8	60.3	75.3	48.9	34,867	2.87	7.9	5.7	
3	×	✓	×	87.7	69.1	75.7	59.9	71.8	48.2	23,031	3.18	8.5	6.3	
4	×	×	✓	86.1	68.7	75.2	61.2	71.8	48.4	14,152	2.56	6.9	5.1	
5	✓	✓	×	92.2	67.4	76.2	61.0	71.9	46.8	14,730	3.04	8.2	6.1	
6	✓	×	✓	90.2	66.5	76.1	62.1	74.6	48.8	21,987	2.42	6.6	4.9	
7	×	✓	✓	91.3	68.7	77.3	61.7	74.5	46.8	36,430	2.73	7.2	5.5	
8	✓	✓	✓	92.1	70.5	77.7	62.8	75.0	45.6	19,558	2.59	6.9	5.2	

Figure 9 YOLOv8n-Coal performance metrics.

The analysis demonstrates that incorporating the DCNv3 module enhances the model’s ability to accurately determine the target position. This improvement is reflected in the metrics: AP50, AP50:95, and AR50:95 show enhancements of 0.1%, 0.8%, and 1.7% respectively. Additionally, the inclusion of the DCNv3 module results in a reduction of 0.14M parameters, 0.3G FLOPs, and 0.3 MB model weights. Consequently, this reduction in parameters and computational requirements maintains a slight increase in accuracy. The PSA module enhances the model’s performance by eliminating the influence of irrelevant factors. It achieves a 1.0% improvement in AP50 and a 0.4% improvement in AP50:95. However, it slightly reduces the AR50:95 by 1.8%. Additionally, it improves the number of parameters, FLOPs, and model weights by 0.17M, 0.3G, and 0.3 MB, respectively. These improvements have a minimal impact on parameter counts and computation, but significantly enhance the model’s detection precision. The slight change in average recall may be attributed to the inherent randomness in the deep learning training process. The C2fGhost module has a more efficient design, resulting in a 0.5% improvement in AP50 and a 1.7% improvement in AP50:95. However, the AR50:95 is slightly reduced by 1.8%. Additionally, the number of parameters, FLOPs, and model weights are reduced by 0.45M, 1.3G, and 0.9 MB respectively. These reductions significantly decrease the model's complexity and computational requirements, while still maintaining the algorithm’s accuracy or even slightly improving it.

By simultaneously implementing the three enhanced strategies in the YOLOv8n model, we observe improvements in AP50, AP50:95, and AR50:95 by 3.0%, 3.3%, and 1.4% respectively. Additionally, oLRP is reduced by 1.5% and the count of predicted bounding boxes is decreased by 5,997. Furthermore, the number of simultaneous parameters, FLOPs, and model weight are reduced by 0.42M, 1.3G, and 0.8 MB respectively. The model demonstrated significant enhancements in recognition accuracy, recall, and target localization precision. It also effectively reduced false positives and improved detection efficiency. As a result, deploying the model on end devices became less challenging, highlighting the effectiveness of the improved approach.

Comparison experiments

Comparison experiments are carried out to assess the accuracy and efficiency of various target detection algorithms in coal-rock detection tasks. Well-known target detection algorithms such as Faster R-CNN (Ren et al., 2016), YOLOv3 (Redmon & Farhadi, 2018), YOLOv5n, YOLOv5s, YOLOv7 (Wang, Bochkovskiy & Liao, 2022), YOLOv8n, YOLOv10n, YOLOv10s (Wang et al., 2024) and Gold-YOLOn (Wang et al., 2023b) are chosen for comparison with YOLOv8n-Coal to further validate its performance.

Furthermore, to showcase the efficacy of the method presented in this article and the applicability of the proposed module to other models, we have chosen to modify YOLOv5n/s and YOLOv10n/s by integrating the enhanced module described in this article. The enhanced models are denoted as YOLOv5n/s-Coal and YOLOv10n/s-Coal, respectively. The model is adjusted using the same methodology as YOLOv8-Coal. Specifically, for YOLOv5, the C3 modules in the 2nd, 4th, 6th, and 8th layers of the backbone network were replaced with C2fGhost modules, and the C3 modules in the 13th, 17th, 20th, and 23rd layers of the feature fusion network were replaced with DCNv3 modules. Finally, PSA modules were added following the DCNv3 modules located just before the three detection heads, at layers 18, 22, and 26. Similarly, for YOLOv10, the C2f and C2fCIB modules in the 2nd, 4th, 6th, and 8th layers of the backbone network were replaced with C2fGhost modules, and the C2f and C2fCIB modules in the 13th, 16th, 19th, and 22nd layers of the feature fusion network were replaced with DCNv3 modules. Lastly, PSA modules were added following the DCNv3 modules located before the three detection heads, at layers 17, 21, and 25. By modifying YOLOv5 and YOLOv10, this series of experiments aims to demonstrate how the improved modules function in different models, thereby confirming the effectiveness of this method across various model architectures.

The datasets are uniform throughout all experiments, and same parameters are utilized. Figure 10 displays the Precision-Recall curves for various algorithms, while Table 3 presents the experimental results.

Figure 10 Precision-Recall curves for different algorithms.

Table 3 Results of the comparison experiments.

Bold entries represent the highest values for each metric.

Model	Precision
(%)	Recall
(%)	AP50
(%)	AP50:95
(%)	AR50:95
(%)	oLRP
(%)	BBox count	Params
(M)	FLOPs
(G)	Weight
(MB)	
Faster R-CNN	56.5	75.3	75.0	51.6	61.4	53.5	5,999	137.10	370.2	522.9	
YOLOv3	88.7	65.4	73.2	57.0	73.2	53.9	27,443	61.52	155.3	117.8	
YOLOv5n	82.4	61.0	69.0	51.5	72.2	57.9	15,007	1.77	4.2	3.7	
YOLOv5n-Coal	88.3	66.9	73.4	57.9	73.7	51.6	20,814	1.84	4.1	3.7	
YOLOv5s	87.9	62.0	70.4	55.0	67.3	54.8	18,140	7.02	15.9	13.8	
YOLOv5s-Coal	90.3	66.9	75.3	60.1	72.2	49.8	14,329	7.29	15.4	13.8	
YOLOv7	91.2	66.0	73.8	57.1	66.8	51.4	2,302	37.20	105.1	71.3	
YOLOv8n	90.7	68.2	74.7	59.5	73.6	47.1	25,555	3.01	8.2	6.0	
YOLOv10n	89.4	68.7	75.2	57.7	72.6	50.5	143,100	2.30	6.7	5.50	
YOLOv10n-Coal	91.2	70.7	76.1	60.5	74.7	48.3	143,100	2.06	5.4	5.15	
YOLOv10s	88.2	67.2	73.8	57.7	76.2	51.3	143,100	7.25	21.6	15.76	
YOLOv10s-Coal	89.7	69.6	76.1	61.3	76.4	49.4	143,100	7.28	17.4	15.88	
Gold-YOLOn	94.0	67.7	73.6	59.2	67.7	45.5	7,121	5.60	12.05	12.0	
YOLOv8n-Coal (Ours)	92.1	70.5	77.7	62.8	75.0	45.6	19,558	2.59	6.9	5.2	

YOLOv8n, as the baseline model, surpasses the majority of models in the task of recognizing coal rocks. While Faster R-CNN may have a slightly higher AP50, YOLOv8n stands out for its notable lightweight features, with much fewer parameters and FLOPs compared to Faster R-CNN. Hence, enhancing the performance of YOLOv8n while preserving its lightweight advantage can be achieved by choosing to improve it. The enhanced YOLOv8n-Coal shows significant improvement and exhibits exceptional performance across several measures, with AP50 and AP50:95 values of 77.7% and 62.8%, respectively, which are the best results obtained in the studies. Furthermore, YOLOv8n-Coal has a significantly high AR50:95 of 75.0%, with an OLRP value that is only 0.1% greater than that of Gold-YOLOn, which stands at 45.6%. This enhanced model demonstrates its capacity to be effective and feasible in application situations with limited resources.

In the model improvement experiments mentioned earlier, both the YOLOv5 and YOLOv10 series exhibit substantial enhancements in performance. The inclusion of the three modules in the YOLOv5 series leads to enhancements in the performance metrics of AP50, AP50:95, and AR50:95 for both YOLOv5n and YOLOv5s. Additionally, the decrease in the oLRP metrics indicates a significant reduction in the rate of misdetection. Despite the DCNv3 module having more parameters than the C3 module, resulting in an increase in the total number of parameters in the model, the optimized network structure manages to reduce overall computation.

Similarly, the YOLOv10 series exhibits a comparable pattern of enhancing performance. Based on the experimental findings, the YOLOv10s demonstrates inferior performance compared to the smaller YOLOv10n in crucial performance metrics such as AP50 and oLRP. This discrepancy can be attributed to the integration of the Compact Inverted Block CSP Bottleneck with two Convolutions (C2fCIB) module in the backbone network of YOLOv10s. This module seeks to improve feature characterisation by employing deep feature iteration. However, it also results in increased model complexity and leads to overfitting, particularly when applied to the coal-rock dataset. Following the implementation of the proposed approach in this study, both YOLOv10n and YOLOv10s demonstrate enhanced performance and a decrease in the oLRP value. This not only illustrates the notable impact of the enhanced module in enhancing accuracy, but also provides additional evidence supporting the notion that the C2fCIB module may have caused a decline in the performance of the YOLOv10s. (The YOLOv10s-Coal replaces the C2fCIB module with the C2fGhost module.) The experimental results not only confirm the efficacy of the new module in various target detection networks, but also highlight its crucial role in enhancing network performance.

To summarize, incorporating the DCNv3 module, PSA attention mechanism, and C2fGhost lightweight module into YOLOv8n can significantly enhance the precision of coal-rock identification. Furthermore, the approach presented in this article successfully enhances the accuracy of various models while maintaining a consistent number of parameters and reducing computational requirements. Notably, the most substantial impact is observed on the YOLOv8 model. This is primarily due to the inherent adaptability of the YOLOv8 architecture for feature extraction and speed optimisation, allowing for seamless integration and deeper optimisation of the added modules within the existing architecture. Hence, the incorporation of these modules into YOLOv8n is imperative, thereby reinforcing the need and benefits of the enhanced approach put up in this research.

Visualization of experimental results

To confirm the practical application effect of the model, the YOLOv8n and YOLOv8n-Coal models were chosen to compare the detection results of coal-rock images, as depicted in Fig. 11. Group A observed a decrease in misidentifying rocks as coals and a reduction in incorrect detections with the improved model. Group B noted an enhanced ability of the model to identify more complete coal mines. Group C found that the improved model excelled in detecting coal mines even in cases of worker occlusion, while decreasing false detections. Groups D and E reported that the improved model adhered better to the data labeling strategy, disregarded challenging aspects like low light and blurry images, avoided learning unstable features, and enhanced the model’s reliability. Overall, the YOLOv8n-Coal model demonstrates superior recognition accuracy and effectively addresses issues related to imprecise coal target positioning.

Figure 11 Comparison of detection results.

Image credit: Wang et al. (2024). Coal and Rock [Data set]. Zenodo. https://doi.org/10.5281/zenodo.10702704.

Conclusions

A YOLOv8-based model named YOLOv8-Coal is introduced in this article to overcome the limitations faced by the conventional YOLOv8 in detecting coals and rocks. The DCNv3 module is introduced to adjust to variations in the shape and size of the coals target, enhancing algorithm accuracy. The PSA module is integrated into the feature fusion network to maximize important image features and enhance model accuracy. The C2fGhost module is utilized to optimize the balance between accuracy and computational efficiency. Experiments demonstrate that YOLOv8-Coal has superior recognition accuracy for coal and rock images. It also enhances model lightness, aiding in detecting coal mine distribution and improving operational efficiency on underground equipment. This offers a more dependable and efficient technical solution for unmanned operations in coal mines. Future research will focus on enhancing the operational efficiency and real-time performance of the model through hardware co-design to address the pressing demand for high-precision and high-efficiency recognition technology in the mining industry.

Supplemental Information

Supplemental Information 1 Literature table for coal-rock image recognition.

Additional Information and Declarations

Competing Interests

Author Contributions

Data Availability

The authors declare that they have no competing interests.

Wenyu Wang conceived and designed the experiments, performed the experiments, analyzed the data, performed the computation work, prepared figures and/or tables, authored or reviewed drafts of the article, and approved the final draft.

Yanqin Zhao conceived and designed the experiments, analyzed the data, authored or reviewed drafts of the article, and approved the final draft.

Zhi Xue performed the experiments, prepared figures and/or tables, and approved the final draft.

The following information was supplied regarding data availability:

The literature table for coal-rock image recognition is available in the Supplemental File.

The code of YOLOv8-Coal is available at Zenodo: Wang, W. (2024). Jason-2k/YOLOv8-Coal: V 1.1 (release1.1). Zenodo. https://doi.org/10.5281/zenodo.12582996.

The model weights are available at Zenodo: Wang, W. (2024). The model weights file of YOLOv8-Coal. Zenodo. https://doi.org/10.5281/zenodo.10728156.

The code of YOLOv5-Coal and YOLOv10-Coal are available at Zenodo:

Wang, W., & Xue, Z. (2024). Jason-2k/YOLOv8-Coal-Comparison-experiments: V 1.0 (release). Zenodo. https://doi.org/10.5281/zenodo.12704209

The original dataset is available at Zenodo: Wang, W. (2024). Coal and Rock [Data set]. Zenodo. https://doi.org/10.5281/zenodo.10702704

The processed dataset is available at Zenodo: Wang, W. (2024). Coal and Rock after processing in YOLO format [Data set]. Zenodo. https://doi.org/10.5281/zenodo.10702879.

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
