# Peer review of "YOLOv8-Coal: a coal-rock image recognition method based on improved YOLOv8"

_PeerJ Computer Science, doi:10.7717/peerj-cs.2313_

## Round 0.1 · original submission · Major Revisions

Please revise the paper according to the comments of the 2 reviewers.

Reviewer 1 ·

Basic reporting

Professional article structure, figures, tables.

Experimental design

Well defined resarch objective

Validity of the findings

Clear methodology:
Reliable data collection:

Additional comments

Confounding variables may lead to unpredictable outcome

Cite this review as

Reviewer 2 ·

Basic reporting

The references are relevant adequate and sufficient.
Technical writing is good
Has some readership appeal.

Experimental design

The results are bit good
Added value to the domain

Validity of the findings

Results and conclusion are good

Additional comments

Has technical merit and readership appeal.

Cite this review as

Reviewer 3 ·

Basic reporting

no comment.

Experimental design

no comment.

Validity of the findings

No comment.

Additional comments

In this paper, the authors have used a new method introduced to enhance recognition accuracy and processing speed in odrder to address issues such as misdetection and omission due to low light, image defocus, and worker occlusion in coal-rock image recognition. Extensive experimental results demonstrate the efficiency of the proposed method. Overall, the issue of this study is of practical significance. However, there are a few points that need further improvement.
1. In the Introduction, more relevant references could be cited to analyse the relevant work.
2. In the Materials & Methods, the authors should highlight the innovative work in this paper, instead of simply listing the modular frameworks of other reference.
3. The figure 3 of Deformable Convolutional Networks Version 3 is the same as Deformable Convolutional Networks Version 1, please give a more detailed explanation.
4. Parameters in the equations need to be specifically given an explanation, for example, in Eq(4), ech and esp should be explaned.
5. Image resolution is too low to see specific information, which needs to improve image clarity.
6. Suggested modifications to Table 2 for ablation experiments for which a specific table template exists.
7. Further experiments are needed to explain why the module introduced in the paper must be used onYOLOv8 and how it works on other YOLOversions.

Cite this review as

Reviewer 4 ·

Basic reporting

All comments have been added in detail to the 4th section called additional comments.

Experimental design

All comments have been added in detail to the 4th section called additional comments.

Validity of the findings

All comments have been added in detail to the 4th section called additional comments.

Additional comments

Review Report for PeerJ Computer Science
(YOLOv8-Coal: A coal-rock image recognition method based on improved YOLOv8)

1. Within the scope of the study, a YOLOv8-based deep learning model specific to the study was developed to perform object detection operations in coal-rock images.

2. In the Introduction section, the definition of coal, coal-rock image recognition technologies, studies in the literature and the main contributions of the study are mentioned. In order to emphasize the difference and importance of the study from the literature, it is recommended to add a literature table consisting of columns such as "dataset, advantage, disadvantage, data preprocessing/augmentation methods, results, detection models" to this section. Thus, the place of this study in the literature will gain more importance.

3. In the study, it was stated that the dataset consisting of coal-rock images was collected in high resolution from two different coal mines specific to the study. The fact that the dataset is specific to the study rather than open source increases the value of the study and its originality in terms of the dataset.

4. It was stated that the dataset distribution was randomly selected as 70%, 20% and 10% for training, validation and testing, respectively. Since object detection results are very dependent on the dataset, how the dataset is distributed and its reasons are very important. Normally, cross-validation can be performed to obtain more accurate detection results. Explain in detail the reason why the dataset distribution was chosen as 70%, 20%, 10% within the scope of this study and/or the reason why cross-validation was not preferred.

5. It was stated that initially a small amount of the dataset was divided into pieces at high resolution and then augmentation operations were performed. I recommend that the dataset be presented in a table, including the initial state, the state after being divided into pieces, the state after augmentation, and each dataset part (training, validation, test) for the change in the number of objects (coal target) in all these states. In terms of the Object class, it should be clarified more clearly whether there is only one type or more than one type.

6. In the Experiment section, the hardware used and the basic program section are included. At this stage, more detailed information such as the framework/toolbox used within the scope of the study should be given.

7. It should be explained more clearly how the parameters such as epoch, optimizer and learning rate given in Table-1 were determined and whether other values were tried.

8. When the YOLOv8-Coal network structure proposed in the study is examined in detail and compared with the literature, it is observed that the modules and their originality are at a sufficient level.

9. For accurate analysis of the results, the evaluation metrics required to solve object detection problems must be obtained completely. For this reason, it is recommended to obtain missing metrics such as Average Recall (AR), Optimal Localization Recall Precision (oLRP) and count of predicted bounding box for all compared models, including the proposed model with and without augmentation.

10. In the study, the results obtained with the proposed YOLO-based models, different versions of YOLO models and Faster R-CNN were compared. When we look at single-stage and two-stage object detectors, it is recommended to compare the results of several current new object detectors in terms of the depth of the study.

11. For images obtained from both coal mines, it is recommended to add sample images that include both ground-truth bounding boxes and predicted bounding boxes.

As a result, the study is of a certain quality in terms of both the dataset and the proposed object detection model, but it should be detailed in terms of the parts mentioned above.

Cite this review as

---

## Round 0.2 · accepted · Accept

Thanks to the authors for their efforts to improve the work. The current version successfully satisfied the reviewer. It may be accepted now. Congrats to the authors!

Reviewer 4 ·

Basic reporting

All comments have been added in detail to the last section.

Experimental design

All comments have been added in detail to the last section.

Validity of the findings

All comments have been added in detail to the last section.

Additional comments

Review Report for PeerJ Computer Science
(YOLOv8-Coal: A coal-rock image recognition method based on improved YOLOv8)

Thanks for the revision. When both the detailed responses to the reviewer comments and the relevant changes made to the paper are examined in detail, it is observed that they are sufficient. Since this YOLO-based study has the potential to make a very important contribution to the literature, I recommend that the paper be accepted. I wish the authors success in their future work. Best regards.

Cite this review as